# Semantic-Guided Hierarchical Stackelberg Games for Multi-Agent Coordination with Fuzzy Constraints

## Abstract

Multi-agent coordination under complex constraints remains a fundamental challenge in distributed optimization, particularly when real-world problems involve constraints that practitioners describe informally. While humans intuitively understand and balance such requirements across multiple decision-makers, automated systems demand precise mathematical formulations that may not exist or may lose essential context in translation. We propose a framework where distributed agents learn to interpret and satisfy semantic constraints through collective intelligence rather than explicit programming. The key insight is that natural language constraints often encode contextual priorities that vary with system state, which we capture by treating constraints as adaptive geometric structures whose shape depends on operational context. When an operator specifies "maintain reasonable reserves," the meaning of "reasonable" naturally differs during peak versus off-peak periods, and our framework learns these contextual variations through language model guidance. To coordinate multiple agents under such fluid constraints, we structure the problem as a hierarchical game where semantic understanding at the system level guides local agent decisions without imposing rigid rules, allowing emergent coordination through game-theoretic interactions. The framework remains computationally tractable by recognizing that different aspects of coordination occur at different timescales, with fast local adjustments separated from slower global consensus formation through singular perturbation decomposition. Empirical evaluation on a multi-agent resource allocation task demonstrates that semantic-aware coordination achieves 89.2% efficiency compared to 82.7-86.4% for traditional methods while successfully handling requirements that resist mathematical formalization.

## 1 Introduction

The energy system of lunar rovers, as mission-critical equipment, directly determines the success or failure of deep space exploration missions(Lei et al., 2025). The harsh lunar environment renders this challenge exceptionally severe: the 300°C day-night temperature differential not only causes dramatic fluctuations in battery chemical reaction efficiency but also combines with the 14-day polar night environment to force the system to independently sustain the rover's vital operations when solar power is completely unavailable(Sun et al., 2024; He et al., 2025). When rovers enter permanently shadowed craters in regions such as the lunar south pole, these areas that perpetually lack direct sunlight maintain extremely low temperatures, causing battery efficiency to decline precipitously and subjecting the system to energy consumption challenges more severe than conventional polar nights(Liu et al., 2023). The absence of atmospheric protection subjects electronic components to continuous cosmic ray bombardment, further exacerbating system design complexity under stringent mass constraints(Rahman et al., 2020), while low-temperature-induced battery efficiency degradation renders every joule of energy precious (Leita & Bozzini, 2024). Given the reality of being 380,000 kilometres from Earth with no possibility of repair, the system's intelligent allocation strategy must necessarily exhibit distinct hierarchical characteristics: during normal scientific operations, the system must carefully balance multiple objectives including battery life maximization and thermal load equilibrium while ensuring basic power output; however, once the rover encounters

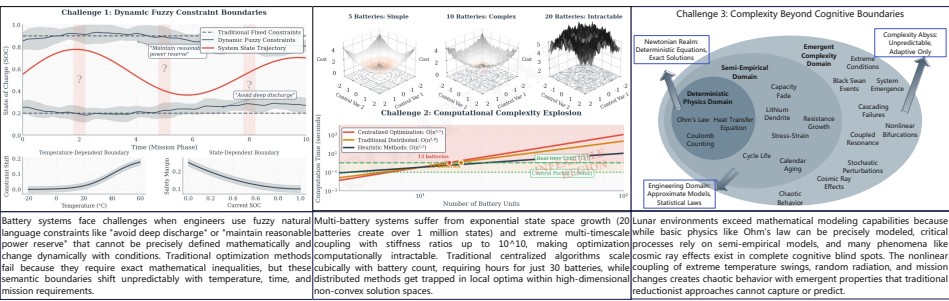

Figure 1: Three core challenges faced by battery systems. Challenge 1: Dynamic fuzzy constraint boundaries demonstrate the difficulty of traditional methods in handling natural language constraints (e.g., "avoid deep discharge") and their temperature-dependent dynamic boundaries; Challenge 2: Computational complexity explosion illustrates the exponential growth of complexity as multi-battery system scale increases; Challenge 3: Complexity beyond cognitive boundaries shows the evolution from simple mathematical modeling to complex systems requiring multi-domain knowledge integration, highlighting the fundamental limitations of traditional reductionist approaches in addressing emergence, uncertainty, and coupling effects.

emergency situations, the system must decisively switch to mission assurance mode, concentrating all available energy to ensure completion of core detection tasks even at the expense of equipment longevity or permanent damage(Shang et al., 2025; Zheng et al., 2025). To achieve this objective, the allocation model requires comprehensive modelling capabilities to address multi-temporal, multi-physical, and multi-objective strongly coupled characteristics, including dynamic modelling capabilities that adapt to parameter time-variance and integrated reasoning capabilities that synthesize interdisciplinary knowledge(Borah et al., 2024; Zou et al., 2025b). However, the absence of these capabilities in existing modelling approaches directly leads to decreased system prediction accuracy and multi-objective coordination failure, subsequently triggering critical issues such as power allocation imbalance, thermal load concentration, and battery life degradation, ultimately threatening the safe execution of the entire exploration mission(Thelen et al., 2024; Esmaeilion & Soltani, 2025).

The essence of these challenges lies in the dynamic uncertainty of the lunar environment causing system constraint boundaries to exhibit high ambiguity, rendering traditional mathematical modelling methods fundamentally limited. During lunar rover operations, the nonlinear evolution of temperature fields, random fluctuations in radiation environments, and sudden changes in mission requirements constitute a constraint space with continuously varying boundaries, where the time-varying characteristics of key parameters exceed the applicable range of static mathematical descriptions(Schwarz et al., 2022; Zou et al., 2025a). More fundamentally, even if system mathematical representations could be constructed through approximation methods, the resulting high-dimensional optimization problems exceed the processing capabilities of existing computational frameworks. The evolution trajectory of system states in complex constraint spaces exhibits strong nonlinearity and high sensitivity characteristics, where minute parameter perturbations may trigger significant changes in system behaviour, while traditional deterministic solution algorithms lose their theoretical foundation in such inherently uncertain environments(Shick et al., 2024; Zhao et al., 2025). Therefore, the current challenge encompasses not only technical issues of modelling accuracy and computational efficiency, but more fundamentally, the cognitive boundaries of mathematical tools when processing dynamic fuzzy constraint systems, constituting a fundamental technical bottleneck in lunar rover battery management system design.

Although existing algorithms are theoretically mature, they still possess fundamental limitations that are difficult to overcome when facing practical applications in energy systems as in Fig 1. These algorithms cannot effectively address the "complexity barrier", with computational complexity growing exponentially with system scale; they struggle to achieve "bounded rationality" decision-making, with severely insufficient capability to find acceptable solutions within limited time; simultaneously, they lack necessary "interpretability", with decision processes completely opaque to operators, making it difficult to establish operational trust(Steyvers & Kumar, 2024). More critically, these methods exhibit extremely poor adaptability—when system parameters change or new constraint conditions

emerge, complete remodelling is often required, which proves particularly fatal when lunar rovers face extreme environmental changes, unable to rapidly respond to dynamically changing practical requirements(Ukoba et al., 2024). Therefore, there is an urgent need for an innovative framework capable of breaking through the fundamental limitations of traditional optimization paradigms.

The innovative combination of large language models and game theory provides a breakthrough solution to this challenge(Ling, 2022). Large language models acknowledge that overall behaviour may not be predictable from simple combinations of parts, requiring higher-level intelligence for co-ordination, and demonstrate powerful cognitive intelligence advantages with natural capabilities for processing incomplete information and fuzzy constraints, enabling cross-domain knowledge integration and rapid learning adaptation, while providing intuitively understandable decision explanations for humans(Wang, 2025). Game theory, particularly Stackelberg games, provides precise coordination mechanism advantages: its hierarchical structure naturally accommodates the hierarchical decision-making requirements of battery allocation, multi-agent game frameworks can effectively handle coordination and competition relationships among batteries, and the mathematical existence of equilibrium solutions guarantees theoretical system stability(Li et al., 2021; Cheng et al., 2025). Large language models specifically handle the "incomputable" complexity components that traditional methods cannot address, while game theory ensures "provable" mathematical convergence, thereby achieving organic unity between cognitive intelligence and mathematical precision. This paradigmatic shift holds significant importance for mission-critical systems operating in extreme environments such as lunar rovers, completing the important paradigmatic transition from "perfect rationality" to "bounded rationality"(Gershman et al., 2015).

In response, we innovatively propose a hierarchical Stackelberg game optimization framework based on semantic embedding. This framework remodels complex system optimization as a hierarchical decision-making problem deeply integrating cognitive intelligence and game theory, achieving intrinsic unity between fuzzy semantic processing and precise game equilibrium by embedding the semantic understanding capability of natural language constraints into the hierarchical decision structure of Stackelberg games, providing an effective strategy for alleviating fundamental challenges such as dynamicity and uncertainty, combinatorial explosion and complexity, and knowledge acquisition bottlenecks. The framework transcends the limitations of traditional reductionism, acknowledging that overall behaviour cannot be predicted from simple combinations of parts, transforms fuzzy constraints into mathematically comprehensible objects for game participants through semantic embedding, and ensures organic unity between individual rationality and system-wide objectives through hierarchical game structures. Cognitive intelligence handles the "incomputable" complexity components, while game theory guarantees "provable" mathematical convergence, synergistically achieving the paradigm transition from "perfect rationality" to "bounded rationality," providing a theoretically rigorous and practically viable unified solution for hierarchical decision-making in multi-agent systems.

## 2 PROBLEM DEFINITION

This chapter establishes a mathematical model for multi-battery cooperative allocation systems, characterizing complex interaction behaviors and coordination decision mechanisms among battery units in new energy power systems. Consider a power system composed of N heterogeneous battery units with different capacities, internal resistances, thermal characteristics, and aging rates. The state space is defined as:

$$\mathcal{X} = \prod_{i=1}^{N} \mathcal{X}_i \subset \mathbb{R}^{n \times N} \tag{1}$$

where $\mathcal{X}_i$ represents the state space of the $i$-th battery, and the state vector $x_i = [SOC_i, T_i, SOH_i, R_i]^T \in \mathcal{X}_i$ characterizes the battery's State of Charge, Temperature, State of Health, and Internal Resistance. The system dynamics exhibit strong coupling characteristics, where individual battery state evolution depends on their own control inputs and the influence of other batteries through thermo-electrical coupling mechanisms:

$$\dot{x}_i = f_i(x_i, u_i, \sum_{j \neq i} \phi_{ij}(x_j, u_j)) \tag{2}$$

where $u_i \in \mathcal{U}_i$ is the control input for the $i$-th battery, and $\phi_{ij}$ characterizes the coupling influence between batteries through electrical and thermal mechanisms.

The optimization objectives require balancing multiple conflicting performance metrics. Instantaneous power tracking accuracy requires total output power to match time-varying demands:

$$J_1 = \mathbb{E}\left[\left\|\sum_{i=1}^{N} u_i - P_{req}(t)\right\|^2\right] \tag{3}$$

Long-term lifetime degradation is quantified through cumulative damage:

$$J_2 = \sum_{i=1}^{N} \int_0^T L_i(x_i, u_i)dt \tag{4}$$

Thermal load balance minimizes temperature distribution variance:

$$J_3 = \text{Var}(\{T_i\}_{i=1}^N) = \frac{1}{N}\sum_{i=1}^{N}(T_i - \bar{T})^2 \tag{5}$$

The constraint space contains deterministic hard constraints reflecting rigid requirements of physical laws and safety boundaries:

$$h(x, u) = 0 \tag{6}$$

and fuzzy soft constraints with semantic uncertainty:

$$g(x, u, \xi) \leq 0 \tag{7}$$

where $\xi \in \Xi$ represents uncertainty in semantic parameter space. The decision architecture employs a two-layer Stackelberg-Nash game framework where the system coordinator designs incentive signals $\lambda$ to guide battery behavior:

$$\min_{\lambda \in \Lambda} F(\lambda, u^*(\lambda)) \tag{8}$$

whilst individual battery agents compute optimal responses:

$$u_i^* \in \arg\min_{u_i \in \mathcal{U}_i} f_i(u_i, u_{-i}, \lambda) \tag{9}$$

The system operates under a Partially Observable Markov Decision Process framework, where each battery agent obtains local information through noisy observations:

$$y_i = h_i(x_i) + v_i \tag{10}$$

where $v_i \sim \mathcal{N}(0, \Sigma_i)$ is Gaussian observation noise. The core challenges are manifested in three aspects: semantic ambiguity and time-variance of constraint boundaries, computational complexity caused by multi-timescale coupling, and the trade-off between sub-optimality and stability in distributed decision-making. These challenges require transcending traditional optimization paradigms and exploring innovative methods that integrate cognitive intelligence with mathematical precision.

## 3 METHODS

### 3.1 TOPOLOGICAL SEMANTIC EMBEDDING OF FUZZY CONSTRAINT SPACES

The constraint space of battery allocation systems exhibits dual characteristics of dynamic evolution and semantic ambiguity, which traditional optimization methods cannot effectively handle. We propose a constraint processing theory that integrates cognitive intelligence with geometric analysis, establishing a unified framework from fuzzy semantics to precise geometric representation through deep integration of large language model semantic understanding capabilities with differential geometry mathematical precision as in Fig 2.

Engineers often use fuzzy constraints difficult to describe precisely mathematically, such as "avoid deep discharge to extend lifespan" or "operate conservatively in high-temperature environments." Our framework utilizes large language model cognitive capabilities to handle semantic constraint

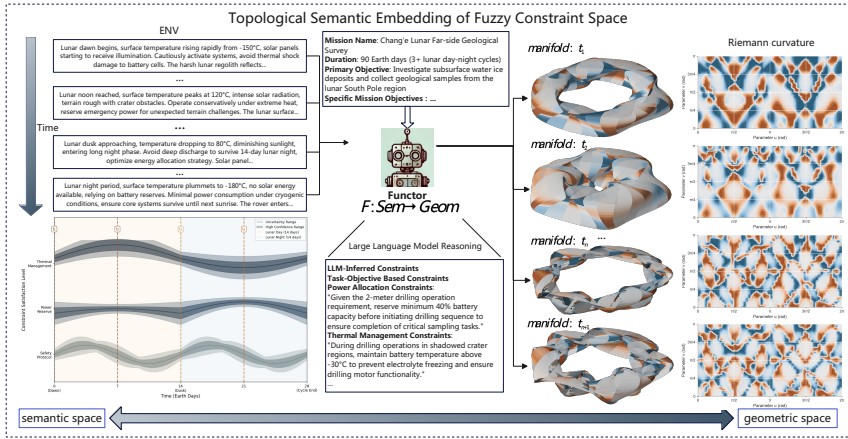

Figure 2: This figure illustrates the topological semantic embedding process of fuzzy constraint spaces, where natural language constraints from lunar exploration missions (such as "reserve minimum 40% battery capacity") are transformed into parameterized differential manifolds through large language model reasoning. This method achieves the mapping transformation from semantic space to geometric space, modeling dynamic constraints as deformable geometric structures with Riemann curvature characteristics, providing a theoretical foundation for the mathematical processing of complex constraints.

understanding, then transforms them into parameterized differential manifolds. The semantic understanding process begins with vectorized representation where constraint description $s = \{w_1, w_2, ..., w_L\}$ is mapped to high-dimensional latent space:

$$z = \mathcal{E}_\theta(s, c) \in \mathbb{R}^d \tag{11}$$

where $c$ is contextual battery system information. We learn a state-dependent Riemannian metric:

$$d_{\mathcal{H}}(z_1, z_2 | x) = \sqrt{(z_1 - z_2)^T G(x)(z_1 - z_2)} \tag{12}$$

reflecting that identical constraint descriptions have different urgency under different operating conditions. Cross-modal alignment achieves effective mapping from semantics to mathematics through Wasserstein optimal transport theory. The time-varying constraint set is modeled as a parameterized differential manifold:

$$\mathcal{M}(t) = \{x \in \mathcal{X} : \Phi(x, t) = 0\} \tag{13}$$

where $\Phi$ contains both deterministic physical constraints and semantically processed fuzzy constraints. Geometric properties including curvature and topological classification provide guidance for subsequent optimization algorithms, whilst hierarchical constraint decomposition through fiber bundle structures enables computational complexity reduction.

## 3.2 Variational Inequality Solution for Two-Layer Stackelberg Game Equilibrium

The hierarchical decision structure naturally fits the Stackelberg game framework: the system dispatcher acts as leader formulating global incentive strategies, whilst individual battery management units act as followers responding to incentives. Each battery i has decision variable $u_i \in \mathcal{U}_i$ with cost function including aging, thermal deviation, and system coordination terms as in Fig 3.

The equilibrium condition leads to a variational inequality problem: find $u^* \in \mathcal{U}$ such that:

$$\langle F(u^*, \lambda), u - u^* \rangle \geq 0, \quad \forall u \in \mathcal{U} \tag{14}$$

where $F_i(u, \lambda) = \nabla_{u_i} J_i(u_i, u_{-i}, \lambda)$. Strong monotonicity of mapping $F$ ensures existence and uniqueness of solutions through battery internal resistance and thermal capacity characteristics. Augmented Lagrangian relaxation addresses constraint qualification failures:

$$\mathcal{L}_\rho(u, \lambda, \mu) = \sum_{i=1}^N J_i(u_i, u_{-i}, \lambda) + \mu^T h(u) + \frac{\rho}{2} \|h(u)\|^2 \tag{15}$$

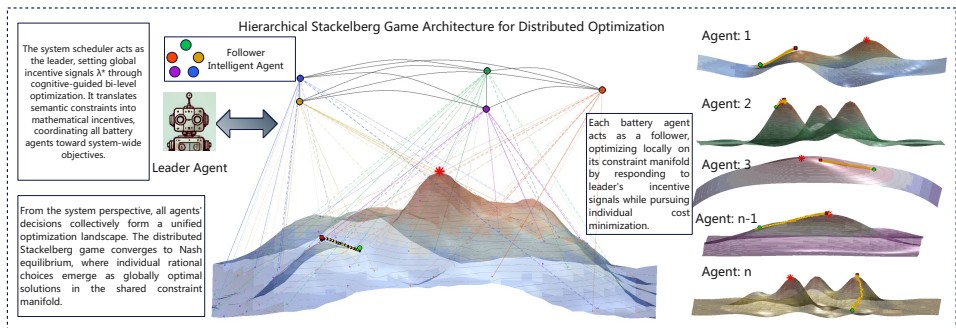

Figure 3: This shows a hierarchical Stackelberg game for distributed battery optimization. The system scheduler (leader) sets global incentive signals, while battery agents (followers) minimize local costs within their constraints. Individual rational decisions collectively converge to a globally optimal Nash equilibrium.

Non-smoothness of battery aging models is handled through subgradient methods, whilst stochastic variational inequality addresses parameter uncertainty. The upper-level Stackelberg leader determines optimal incentive $\lambda^*$ through bi-level optimization, computing gradients via implicit function theorem combined with sensitivity analysis.

## 3.3 COGNITIVE-GUIDED PRIMAL-DUAL DISTRIBUTED ALGORITHM

Actual battery management systems require distributed algorithms enabling individual units to autonomously converge to game equilibrium through local information exchange. The primal-dual framework begins with convex conjugate decomposition as in Fig 4:

$$\min_{u \in \mathcal{U}} \sum_{i=1}^{N} f_i(u_i) + g(Au) \tag{16}$$

Algorithm 1 presents the cognitive-guided distributed solution:

---

**Algorithm 1:** Cognitive-Guided Primal-Dual Distributed Algorithm

---

**Data:** Initial primal state $u^0 = (u_1^0, ..., u_N^0)$, initial dual variable $\lambda^0$, local cost functions $f_i$, global coupling constraints $A$, cognitive model $\mathcal{M}_{\text{cog}}$
**Result:** Optimal primal-dual solution $(u^*, \lambda^*)$
**Initialization**: For each agent $i$, set $u_i^0, \lambda_i^0$, and initial learning rates $\eta_i^0$.
**for** $k = 0, 1, 2, \ldots$ *until convergence* **do**
    **foreach** *agent $i$ in parallel* **do**
        `// Primal update with Mirror Prox`
        Predict optimal momentum from historical trajectories using $\mathcal{M}_{\text{cog}}$.
        Receive dual variables $\lambda_j$ from neighbors $j \in \mathcal{N}(i)$.
        Compute update direction $v_i^k = \nabla f_i(u_i^k) + A_i^T \lambda^k$.
        Update primal variable $u_i^{k+1} = \arg\min_{u_i \in \mathcal{U}_i} \{\langle v_i^k, u_i - u_i^k \rangle + \frac{1}{\eta_i^k} D_{\psi_i}(u_i, u_i^k)\}$.
        `// Dual update`
        Receive primal variables $u_j^{k+1}$ from neighbors $j \in \mathcal{N}(i)$.
        Update dual variable $\lambda^{k+1} = \lambda^k + \gamma(Au^{k+1} - b)$.
        Quantize and communicate $\lambda_i^{k+1}$ to neighbors with error compensation.
        Update learning rate $\eta_i^{k+1} = \mathcal{M}_{\text{cog}}(u_i^{k+1}, \eta_i^k)$.
    **end**
**end**

---

Mirror Prox algorithm based on Bregman divergence provides optimization in non-Euclidean spaces with state-dependent kernel functions $\psi_i(u_i) = \frac{1}{2\eta_i(SOC_i)}u_i^2 + h_i(u_i)$ reflecting charg-

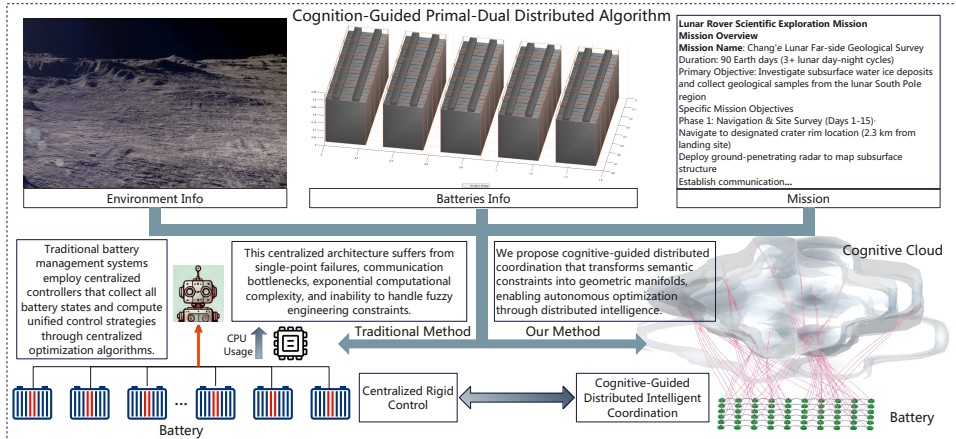

Figure 4: This figure contrasts traditional centralized battery management with the proposed cognition-guided distributed algorithm. The new approach shifts from rigid centralized control to intelligent distributed coordination, using a cognitive cloud to transform constraints into geometric manifolds. This enables autonomous battery optimization while eliminating single-point failures, communication bottlenecks, and constraint handling limitations of centralized systems.

ing/discharging difficulty at different SOC levels. Asynchronous update mechanisms handle communication heterogeneity, allowing each node to update at its own pace with convergence rate:

$$\|u^k - u^*\|^2 \le \frac{C}{\sqrt{k}}(1 + \tau_{max}) \tag{17}$$

Adaptive Heavy-ball momentum accelerates convergence with cognitive models analyzing historical trajectories to predict optimal momentum coefficients. Communication efficiency is improved through intelligent quantization with error compensation mechanisms, whilst Byzantine fault tolerance ensures robustness against node failures through median-based robust aggregation. This distributed framework enables each battery to function as an intelligent agent with autonomous decision-making capabilities, achieving global optimal behavior through local interactions.

## 4 EXPERIMENTS

### 4.1 SIMULATION SYSTEM MODELING

To validate the effectiveness of the proposed semantic embedding-based hierarchical Stackelberg game optimization framework, this study constructs a high-fidelity multi-battery system simulation platform. The platform comprehensively considers the electrochemical characteristics, thermodynamic behavior, aging degradation mechanisms, and system-level constraint coupling of batteries. A third-order equivalent circuit model is adopted to describe battery dynamic characteristics, whilst thermodynamic modeling considers internal heat generation, liquid cooling heat dissipation, heat conduction between batteries, and heat exchange between battery packs. Battery aging modeling adopts a dual-stress model, comprehensively considering calendar aging and cycle aging effects. A 5×4 series-parallel battery pack topology is constructed, totaling 20 battery cells, each equipped with an independent battery management unit. The simulation platform is built on MATLAB/Simulink with modular architecture, using LLAMA as the core large language model for constraint processing and deployed on NVIDIA 3090 GPUs.

### 4.2 EXPERIMENTAL ENVIRONMENT AND EVALUATION METRICS

The experimental environment adopts MATLAB R2023b running on a 64-core AMD EPYC 7742 processor server with 256GB memory. Experimental evaluation focuses on comprehensive optimization performance under different operating conditions, with all experiments repeated 5 times to obtain average values and standard deviations. Key evaluation metrics include energy efficiency

(ratio of useful output work to total consumed energy), average temperature (system average temperature during simulation), battery aging rate (average health degradation rate of all batteries), power tracking error (accuracy in tracking dynamic power demands), temperature overshoot (maximum temperature increase during power steps), settling time (time to reach steady state), steady-state error (long-term tracking accuracy), and robustness index (performance stability under disturbances).

### 4.3 EXPERIMENTAL RESULTS

This chapter validates the effectiveness of the proposed framework through comparative experiments with three baseline algorithms: Model Predictive Control (MPC), Non-dominated Sorting Genetic Algorithm II (NSGA-II), and Deep Q-Network (DQN).

Basic performance comparison under static constraints evaluates optimization performance under standard static conditions with SOC range [0.2, 0.9], temperature upper limit 60°C, constant 50kW power demand for 3600 seconds, and ambient temperature 25°C. Table 1 presents comprehensive performance results. The proposed method achieves 89.2±0.8% energy efficiency, representing improvements of 3.2%, 6.1%, and 7.9% compared to MPC, NSGA-II, and DQN respectively. Average temperature is controlled at 39.1±1.4°C, which is 2.7-5.8°C lower than other methods. Battery aging rate is $2.03\pm0.09\times10^{-6}$/s, representing reductions of 19.4%-35.4% compared to baseline methods.

Table 1: Algorithm Performance Comparison under Static Constraints

| Algorithm | Energy Efficiency (%) | Average Temperature (°C) | Aging Rate ($\times10^{-6}$/s) |
|---|---|---|---|
| MPC | 86.4±1.3 | 41.8±2.1 | 2.52±0.15 |
| NSGA-II | 84.1±1.8 | 43.7±2.8 | 2.81±0.19 |
| DQN | 82.7±2.2 | 44.9±3.2 | 3.14±0.23 |
| **Proposed Method** | **89.2±0.8** | **39.1±1.4** | **2.03±0.09** |

Adaptability comparison under dynamic conditions verifies algorithm performance in complex dynamic environments including power steps, temperature changes, and load fluctuations. Power demand varies in three phases: 0-1200s at 30kW, 1200-2400s at 75kW, and 2400-3600s at 45kW, with ambient temperature varying from 20°C to 35°C then decreasing to 28°C. Table 2 summarizes response performance results. The proposed method's power tracking error is only 3.1±0.5

Table 2: Algorithm Performance Comparison under Dynamic Conditions

| Algorithm | Power Tracking Error (%) | Temperature Overshoot (°C) | Settling Time (s) | Steady-State Error (%) | Robustness Index |
|---|---|---|---|---|---|
| MPC | 4.8±0.7 | 7.6±1.4 | 28.3±3.2 | 2.1±0.4 | 0.73 |
| NSGA-II | 6.9±1.2 | 11.2±2.1 | 45.7±5.8 | 3.4±0.6 | 0.58 |
| DQN | 8.5±1.8 | 13.8±2.9 | 19.6±2.4 | 4.7±0.9 | 0.52 |
| **Proposed Method** | **3.1±0.5** | **4.9±0.9** | **18.2±1.8** | **1.3±0.2** | **0.89** |

Algorithm computational complexity comparison evaluates computational efficiency under different battery pack scales (10, 20, 50, and 100 battery units). Table 3 presents computation time results. MPC algorithm's computation time grows cubically, requiring 3567.1 seconds for 100 batteries, making it difficult to meet real-time requirements. The proposed method takes 412.9 seconds in a 100-battery system, demonstrating good scalability and computational efficiency through multi-scale decomposition and distributed game theory. The experimental results demonstrate that the proposed method achieves superior performance across all key metrics whilst maintaining reasonable computational complexity, validating its effectiveness for practical large-scale battery system applications.

## 5 DISCUSSION

This study comprehensively validates the effectiveness of the proposed method through static constraint and dynamic operating condition experiments. The results demonstrate that the semantic embedding-based hierarchical Stackelberg game framework significantly outperforms traditional methods in key metrics including energy efficiency, temperature control, and battery lifespan. The experiments reveal the framework's excellent adaptability in complex dynamic environments and

Table 3: Algorithm Computation Time Comparison at Different Scales (Unit: seconds)

| Algorithm | 10 Batteries | 20 Batteries | 50 Batteries | 100 Batteries |
|---|---|---|---|---|
| MPC | 28.4±3.2 | 142.6±12.8 | 892.3±78.4 | 3567.1±312.6 |
| NSGA-II | 45.7±5.1 | 78.3±9.6 | 287.6±31.2 | 651.8±58.9 |
| DQN | 8.2±1.1 | 15.8±2.7 | 32.4±4.6 | 71.3±8.2 |
| **Proposed Method** | **21.6±2.8** | **54.7±6.2** | **168.3±18.7** | **412.9±41.3** |

good scalability in large-scale systems, providing strong experimental support for intelligent battery management system applications.

The core advantages of our method stem from profound understanding of complex system essence and paradigmatic breakthroughs. Traditional methods decompose complex systems into simple parts, expecting to predict overall behavior through linear superposition, but this reductionist thinking ignores the nonlinear interactions and emergent effects prevalent in battery systems. Our framework acknowledges that overall behavior cannot be predicted from simple combinations of parts, using the cognitive intelligence of large language models to handle "incomputable" complexity components while game theory ensures "provable" mathematical convergence, achieving the important paradigm shift from "perfect rationality" to "bounded rationality." The topological semantic embedding mechanism enables the system to understand and process fuzzy constraints that are difficult to describe with traditional mathematical models, such as "avoid deep discharge" or "operate conservatively in high-temperature environments," allowing important knowledge from engineering experience to be effectively integrated into the optimization process. The variational inequality-based two-layer game structure ensures the achievement of overall system objectives while respecting the individual characteristics and autonomy of each battery unit, achieving optimal allocation of distributed resources through the cognitive-guided primal-dual distributed algorithm.

From a scientific research value perspective, this study provides a new paradigm for complex system optimization that integrates cognitive intelligence with mathematical precision, breaking through the epistemological limitations of traditional reductionist modeling. The framework demonstrates how to organically combine the semantic understanding capabilities of large language models with the mathematical rigor of Stackelberg game theory, providing systematic solutions for handling complex optimization problems with fuzzy constraints and multi-agent coordination. From an industrial value perspective, the framework directly addresses the dilemma of traditional optimization methods in handling fuzzy operational constraints common in engineering practice, enabling the system to better integrate with existing engineering practices and providing feasible technical pathways for intelligent upgrades in high-tech industries.

Although the theoretical framework proposed in this paper provides new approaches for handling fuzzy constraints and multi-scale optimization in battery systems, there are still some theoretical assumptions that require further refinement. Most critically is the fragility of strong monotonicity conditions in variational inequality formulations—when battery internal resistance is very small or under small current operating conditions, the strong monotonicity may fail, affecting the uniqueness of game equilibrium and algorithm convergence. Future research needs to explore robust algorithm design under weak monotonicity conditions and adaptive constraint manifold construction methods to achieve better balance between theoretical rigor and engineering practicality.

# 6 CONCLUSION

This study innovatively proposes a semantic embedding-based hierarchical Stackelberg game optimization framework that breaks through the limitations of traditional reductionist modeling and achieves intelligent cooperative allocation of multi-battery systems by integrating the cognitive intelligence of large language models with the mathematical precision of game theory. The core contributions of the framework include topological semantic embedding of fuzzy constraint spaces, variational inequality solution for two-layer Stackelberg game equilibrium, and cognitive-guided primal-dual distributed algorithm.

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
