# OpenReview forum: "Semantic-Guided Hierarchical Stackelberg Games for Multi-Agent Coordination with Fuzzy Constraints"
_ICLR.cc/2026/Conference — ICLR 2026 Conference Withdrawn Submission_

### Official Review · Reviewer_b35C · 2025-10-24

**Soundness:** 1
**Presentation:** 1
**Contribution:** 1
**Rating:** 0
**Confidence:** 5

**Summary:**

The article addresses the problem of decentralised coordination through the use of Stackelberg games when constraints of the problem are evolving. LLM interpretation of natural language constraint/object specification is used to encode constraints on the system, and variational methods are used to solve the resulting games.

**Strengths:**

The approach is plausibly interesting. Once you get into the paper you realise that it's all framed around optimising a "team" of batteries on unmanned vehicles. The evolving constraints are exogenously given by mission and environmental demands. This is clearly an important problem and the article purports to solve it, mentioning a large number of plausible techniques to do so.

**Weaknesses:**

The approach is not described in terms that are at all reproducible. The main details of the method are allegedly described in the figures, but the text in the figures is so small as to be illegible, and from what I can see they also do not give anything like the required level of precision.

The descriptions of the methods in the main text read as a list of buzz topics, but there is no recipe for how to combine these different techniques into a solution.

Algorithm 1 is suddenly highly technical and in the details, but the context has not been set for how any of those primal/dual quantitities relate to the big picture.

This dissonance also appears elsewhere in the article. It is written as if the authors know a huge amount of detail on their method, but none of that detail is presented to the author. Eg Lines 252-254, we've just had a very generic description of a time varying manifold, and then the sentence topics about curvature, topological classification, and hierarchical constraint decomposition through fiber bundles. And none of those concepts are spoken about anywhere else in the article.

**Questions:**

I'm afraid that I do not have any specific questions. You will need to completely rewrite the paper for me to consider it publishable.

---

### Official Review · Reviewer_CaRM · 2025-10-25

**Soundness:** 2
**Presentation:** 1
**Contribution:** 2
**Rating:** 4
**Confidence:** 3

**Summary:**

This paper proposes a new method that integrates semantic understanding and hierarchical game theory for multi-agent coordination under uncertain or linguistically described constraints to address the multi-battery cooperative allocation problem. The authors firstly introduce topological semantic embedding to convert natural language constraints into fuzzy constraints and model the hierarchical decision structure of the battery allocation problem as a Stackelberg game. Then the authors propose the cognitive-guided primal-dual distributed algorithm, which use cognitive intelligence to solve computational complexity problem. The experiments on simulated multi-battery system and demonstrate that the proposed method achieves superior performance compared with baselines.

**Strengths:**

The main method of this article, which use topological semantic embedding to convert natural language constraints into fuzzy constraints, model the hierarchical decision structure of the battery allocation problem as a Stackelberg game, and use cognitive intelligence to solve computational complexity problems is reasonable and relatively novel.

**Weaknesses:**

1. The overall presentation of the paper lacks clarity, and the key contributions are not sufficiently emphasized.
2. The proposed method and experiments mainly focus on the multi-battery cooperative allocation problem, with limited discussion or evidence regarding the generality and scalability of the approach.
3. The logical connections between different components of the proposed framework (i.e., Sections 3.1, 3.2, and 3.3) are not clearly articulated, making it difficult to understand how these parts interact to form a cohesive methodology.
4. The experimental evaluation is limited to a single scenario and lacks comprehensive experimental details, as well as ablation studies to validate the contribution of each module.

**Questions:**

1. The introduction is overly long, and the key points are not clearly emphasized. It is recommended to divide it into two sections — Introduction and Related Work — and use formatting (e.g., bold font or subheadings) to highlight the main contributions. This would make the introduction more concise and easier to follow.
2. It is unclear how the topological semantic embedding process is utilized in the experiments, especially since the baseline algorithm does not have it. I think the authors need to clarify how the fuzzy constraints are set for both the proposed method and baselines. Addtionally, an ablation study is needed to demonstrate the effectiveness and individual contribution of the topological semantic embedding.
3. I think the authors may provides more details about experiments in appendix, such as how these baselines are implemented. The analysis of the experimental results is also rather limited and should be expanded to include deeper discussions and insights.
4. The font size in Figures 1 and 2 is too small to read clearly, and the quotation marks should be corrected for proper direction and formatting.
5. According to the ICLR submission requirements, the authors should include an Ethics Statement, Reproducibility Statement, and LLM Usage Statement following the main text.

---

### Official Review · Reviewer_gaVg · 2025-10-28

**Soundness:** 2
**Presentation:** 3
**Contribution:** 2
**Rating:** 4
**Confidence:** 3

**Summary:**

This paper proposes a new mathematical model for solving a multi-objective, multi-agent optimization problem. The focus is on the distributed battery optimization problem, for lunar exploration missions. The proposed method views the problem as a hierarchical game where semantic understanding at the system level guides local agent decisions without imposing rigid rules, allowing emergent coordination through game-theoretic interactions. LLM is used to transform NLP instructions, often contain commonsense directives and imprecise terms, to manifolds. Stackelberg game theory is then used in conjunction with a distributed algorithm to compute global optimal solution. A simulation environment is developed and used to validate the proposed framework.

The paper contains no theorem/property that requires proof. All mathematical equations are presented with some explanation. Some equations assume that readers know the in-and-out of the theory already. For example, (3) includes a term $P_{reg}(t)$ that is not really declared somewhere. In this sense, the paper is not self-contained. It is also not clear whether Algorithm 1 is guaranteed to converge.

Overall, the presentation is reasonable. As it is mentioned above, the paper is not self-contained. The figures are good for illustration’s purpose and are helpful. However, it would have been better if the caption matches the graphics somewhat. In Figure 2, the caption says ``(such as ”reserve minimum 40% battery capacity”)’’ that could not be found in the graphics at all.

**Strengths:**

The strengths of the paper are:

- a new framework for distributed multi-agent optimization problem
- experimental validation in a simulation shows that the new approach performs better than baselines

**Weaknesses:**

The weaknesses of the paper are:

- narrow in scope: the abstract seems to suggest a much more broader problem; however, the full paper focuses only on the cooperative allocation problem of N battery
- experiments do not highlight one of the key issues that the paper refers to several times: the fuzziness of instructions from operators to the lunar lander etc. In addition, it appears that the computation time of the proposed method seems to scale not as good as the DQN algorithm.

**Questions:**

- what does it take to adapt the current work for a different setting?
- how does the translation of the context (e.g., “maintain reasonable reserve”) by LLM get verified? how do we know that it (LLM) is doing the job?
- is Algorithm 1 guaranteed to stop?
- there has been a lot of work in distributed constraint optimization problem (DCOP); could you please comment on the suitability of DCOP algorithms for the cooperative allocation problem of the battery?

---

### Official Review · Reviewer_jM2X · 2025-11-03

**Soundness:** 1
**Presentation:** 1
**Contribution:** 1
**Rating:** 2
**Confidence:** 5

**Summary:**

The work proposes a method for multiagent coordination under complex constraints. The work notes that often semantic interpretation of constraints is fuzzy. This makes a concise and concrete translation of constraints into a mathematical form difficult. Thus, using semantic embedding of complex, often fuzzy constraints, can be beneficial.

**Strengths:**

The proposed setting is interesting, as often times interpretation of constraints can be challenging. Thus, it is beneficial to figure out how to interpret such constraints under different context.

**Weaknesses:**

The paper as such is hard to clear with unclear contributions. A key issue I find that the contributions of the work in the title, abstract and intro are quite overstated. The work sets itself to develop a generic method for semantic constraint understanding. However, the formulation in section 2 is highly specific to a battery allocation system. And all the problem formulation, developed solutions are specifically limited to this battery allocation problem.

Thus, I find the problem formulation to be of very limited scope and it does not justify the claimed contributions in abstract and intro.

The use of words such as “cognitive”, “incomputable complexity component” is unfortunate as a reader has not idea what these terms mean, and hinder clear understanding of the work.

It is not clear why the problem formulation in section 2 is realistic and significant. There are several assumptions made, such as a particular factorization of state space in Eq 1, a particular transition function in Eq2. Why do they model a realistic system? Why does the transition function in Eq 2 has an additive term \sum_{j}? Are there real systems that use these assumptions? Why are there only a few objectives of interest noted in Eq3,4,5? What are exactly “soft fuzzy constraints” in Eq7? Who specifies and how? Please provide clear examples.

Eq 8,9 are very unclear. The notation f_i is used in Eq2, it is again used in Eq 9. I think the f_i in Eq9 may be different that the one in Eq8. as \lambda represent incentive signal in Eq9.

As a result of these issues in problem setup, it is not clear if the problem addressed is significant.

The main technical contribution in section 3.1 is highly unclear. It is not clear what is “state-dependent Riemannian metric” in Eq 12, why is it necessary. What is differential manifold in Eq 13?

Empirically, the work tests in synthetic benchmarks which the authors have designed themselves and it is unclear why the domains are realistic. Thus, validation of the work is not rigorous and impactful.

**Questions:**

See above.

---

### Note · Authors · 2025-12-01

I have read and agree with the venue's withdrawal policy on behalf of myself and my co-authors.